# Bandit Linear Control

**Asaf Cassel**
School of Computer Science
Tel Aviv University
acassel@mail.tau.ac.il

**Tomer Koren**
School of Computer Science
Tel Aviv University
tkoren@tauex.tau.ac.il

## Abstract

We consider the problem of controlling a known linear dynamical system under stochastic noise, adversarially chosen costs, and bandit feedback. Unlike the full feedback setting where the entire cost function is revealed after each decision, here only the cost incurred by the learner is observed. We present a new and efficient algorithm that, for strongly convex and smooth costs, obtains regret that grows with the square root of the time horizon $T$. We also give extensions of this result to general convex, possibly non-smooth costs, and to non-stochastic system noise. A key component of our algorithm is a new technique for addressing bandit optimization of loss functions with memory.

## 1 Introduction

Reinforcement learning studies sequential decision making problems where a learning agent repeatedly interacts with an environment and aims to improve her strategy over time based on the received feedback. One of the most fundamental tradeoffs in reinforcement learning theory is the *exploration vs. exploitation tradeoff*, that arises whenever the learner observes only partial feedback after each of her decisions, thus having to balance between exploring new strategies and exploiting those that are already known to perform well. The most basic and well-studied form of partial feedback is the so-called "bandit" feedback, where the learner only observes the cost of her chosen action on each decision round, while obtaining no information about the performance of other actions.

Traditionally, the environment dynamics in reinforcement learning are modeled as a Markov Decision Process (MDP) with a finite number of possible states and actions. The MDP model has been studied and analyzed in numerous different settings and under various assumptions on the transition parameters, the nature of the reward functions, and the feedback model. Recently, a particular focus has been given to continuous state-action MDPs, and in particular, to a specific family of models in classic control where the state transition function is linear. Concretely, in *linear control* the state evolution follows the linear dynamics:

$$x_{t+1} = A_\star x_t + B_\star u_t + w_t, \tag{1}$$

where $x_t \in \mathbb{R}^{d_x}, u_t \in \mathbb{R}^{d_u}, w_t \in \mathbb{R}^{d_x}$ are respectively the system state, action (control), and noise at round $t$, and $A_\star \in \mathbb{R}^{d_x \times d_x}, B_\star \in \mathbb{R}^{d_x \times d_u}$ are the system parameters. The goal is to minimize the total control costs with respect to cost function $c_t(x, u) : \mathbb{R}^{d_x} \times \mathbb{R}^{d_u} \to \mathbb{R}$ put forward on round $t$.

However, in contrast to the reinforcement learning literature, existing work on learning in linear control largely assumes the full feedback model, where after each decision round the learning agent observes the entire cost function $c_t$ used to assign costs on the same round. In fact, to the best of our knowledge, thus far linear control has not been studied in the bandit setting, even in the special case where the costs are generated stochastically over time.

**Contributions.** In this paper, we introduce and study the *bandit linear control* problem, where a learning agent has to control a known linear dynamical system (as in Eq. (1)) under stochastic noise, adversarially chosen convex cost functions, and bandit feedback. Namely, after each decision round

the learner only observes the incurred cost $c_t(x_t, u_t)$ as feedback. (We still assume, however, that the state evolution is fully observable.) For strongly convex and smooth cost functions, we present an efficient bandit algorithm that achieves $\tilde{O}(\sqrt{T})$ regret over $T$ decision rounds, with a polynomial dependence on the natural problem parameters. This result is optimal up to polylogarithmic factors as it matches the optimal regret rate in the easier stationary (i.e., stateless) strongly convex and smooth bandit optimization setting [26, 17].

The starting point of our algorithmic approach is an approximate reparameterization of the online control problem due to [1, 2], called the Disturbance-Action Policy. In this new parameterization, the control problem is cast in terms of bounded memory convex loss functions, under which the cost of the learner on each round depends explicitly only on her last few decisions rather than on the entire history (this is thanks to strong stability conditions of the learned policies [9]).

As a key technical tool, we develop a new reduction technique for addressing bandit convex optimization with bounded memory. While an analogous reduction has been well established in the full feedback model [3, 1, 2], its adaptation to bandit feedback is far from straightforward. Indeed, loss functions with memory in the bandit setting were previously studied by [4], that showed a black-box reduction via a mini-batching approach that, for an algorithm achieving $O(T^{1/2})$ regret in the no-memory bandit setting, achieves $O(T^{2/3})$ regret with memory. While this technique imposes very few restrictions on the adversary, it degrades performance significantly even for adversaries with bounded memory. In contrast, [3] show that the full-feedback setting enjoys nearly no degradation when the adversary's memory is fixed and bounded. Our new technique establishes a similar lossless reduction for bandit feedback with adversaries restricted to choosing *smooth* cost functions. Combining these ideas with standard techniques in (no-memory) bandit convex optimization [15, 25, 17] gives our main result.

Our techniques readily extend to weakly convex costs with regret scaling as $\tilde{O}(T^{2/3})$ in the smooth case and $\tilde{O}(T^{3/4})$ without smoothness. Moreover, these hold even without the stochastic assumptions on the system noise $w_t$, which were only required in our analysis for preserving the strong convexity of the costs through the reduction to loss functions with memory. We defer further details on these extensions to later sections and choose to focus first on the more challenging case—demonstrating how both the strong convexity and smoothness of the costs are exploited—and where $\tilde{O}(\sqrt{T})$ rates are possible.

**Related work.** The study of linear control has seen renewed interest in recent years. Most closely related to our work are [9, 1, 2], that study online linear control in the full-information setting. The latter paper establishes logarithmic regret bounds for the case where the costs are strongly convex and smooth and the noise is i.i.d. stochastic. Subsequently, [16] established a similar result for fixed and known quadratic losses and adversarial disturbances. Thus, we exhibit a gap between the achievable regret rates in the full- and bandit-feedback cases of our problem. (A similar gap exists in standard online optimization with strongly convex and smooth losses [18, 26].)

A related yet crucially different setting of partial observability was recently studied in [28], that considered the case where the state evolution is not revealed in full to the learner and only a low-dimensional projection of the state vector is observed instead. However, this model assumes full observability of the (convex) loss function following each round, and is therefore not directly comparable to ours. When the underlying linear system is initially unknown (this is the so-called adaptive control setting), regret of order $\sqrt{T}$ was recently shown to be optimal for online linear control even with full feedback and quadratic (strongly convex) costs [7, 27]. Optimal and efficient algorithms matching these lower bounds were developed earlier in [10, 22, 1].

In the reinforcement learning literature on finite Markov Decision Processes (MDPs), regret minimization with bandit feedback was studied extensively (e.g., [23, 19, 11, 13, 5, 24, 20]). Our study can thus be viewed as a first step in an analogous treatment of bandit learning in continuous linear control.

## 2 Preliminaries

### 2.1 Problem Setup: Bandit Linear Control

We consider the setting of controlling a known linear dynamical system with unknown (strongly) convex losses and bandit feedback. The linear system is an instance of the process described in Eq. (1) where $A_\star$ and $B_\star$ are known, initialized for simplicity and without loss of generality at $x_0 = 0$. (Our assumptions on the nature of the various parameters are specified below.) Our goal is to minimize the

total control cost in the following online setting where an *oblivious* adversary chooses cost functions $c_t : \mathbb{R}^{d_x} \times \mathbb{R}^{d_u} \to \mathbb{R}$ for $t \geq 1$. At round $t$:

(1) The player chooses control $u_t$;
(2) The system transitions to $x_{t+1}$ according to Eq. (1);
(3) The player observes the new state $x_{t+1}$ and the incurred cost $c_t(x_t, u_t)$.

The overall cost incurred is $J(T) = \sum_{t=1}^{T} c_t(x_t, u_t)$. We denote by $J_K(T)$ the overall cost of a linear controller $K \in \mathbb{R}^{d_x \times d_u}$, which chooses its actions as $u_t = -Kx_t$. For such controllers, it is useful to define the notion of strong stability [9] (and its refinement due to [2]), which is essentially a quantitative version of classic stability notions in linear control.

**Definition 1** (strong stability). A controller $K$ for the system $(A_\star, B_\star)$ is $(\kappa, \gamma)$−strongly stable $(\kappa \geq 1, 0 < \gamma \leq 1)$ if there exist matrices $Q, L$ such that $A_\star + B_\star K = QLQ^{-1}$, $\|L\| \leq 1 - \gamma$, and $\|K\|, \|Q\|, \|Q^{-1}\| \leq \kappa$. If additionally $L$ is complex and diagonal and $Q$ is complex, then $K$ is diagonal $(\kappa, \gamma)$-strongly stable.

For fixed $\kappa, \gamma$, we define regret with respect to the class $\mathcal{K}$ of $(\kappa, \gamma)$−diagonal strongly stable policies

$$\mathcal{K} = \left\{ K \in \mathbb{R}^{d_x \times d_u} : K \text{ is } (\kappa, \gamma)\text{-diagonal strongly stable w.r.t. } (A_\star, B_\star) \right\}. \tag{2}$$

Beyond its relative simplicity, this class is interesting as it contains an asymptotic global optimum (with respect to *all* policies) when the costs are constrained to a fixed quadratic function, as in classical control. The regret compared to $K \in \mathcal{K}$ is given by $R(T, K) = J(T) - J_K(T)$. The pseudo regret is then defined as

$$\overline{\mathcal{R}}_{\mathcal{A}}(T) = \max_{K \in \mathcal{K}} \mathbf{E}[R_{\mathcal{A}}(T, K)],$$

where the expectation is taken with respect to the randomness of the algorithm, and the system noise.

**Assumptions.** Throughout we assume the following. There are known constants $\kappa_B \geq 1$, and $W, G, C, \alpha, \beta, \sigma > 0$ such that:

1. (System bound) $\|B_\star\| \leq \kappa_B$;
2. (Noise bound) $\|w_t\| \leq W \; \forall t \geq 1$;
3. (Cost bounds) If $\|x\|, \|u\| \leq D$ for large enough $D$,[1] then

$$|c_t(x, u)| \leq CD^2, \quad \|\nabla_x c_t(x, u)\|, \|\nabla_u c_t(x, u)\| \leq GD;$$

4. (Curvature bounds) The costs $c_t(x, u)$ are $\alpha$-strongly convex and $\beta$-smooth;
5. (Noise) The disturbances $w_t$ are independent random variables satisfying $\mathbf{E}[w_t w_t^\mathsf{T}] \succeq \sigma^2 I$.

The above assumptions are fairly standard in recent literature (e.g., [1, 2]).

## 2.2 Online Optimization with Memory

We describe the setting of online optimization with memory [4, 3], which will serve as an intermediate framework for our algorithmic development. In this setting, an oblivious adversary chooses loss functions $f_t : \mathcal{X}_+^H \to \mathbb{R}$ over a domain $\mathcal{X}_+ \subseteq \mathbb{R}^d$, where $H \geq 1$ is the length of the adversary's memory. The game proceeds in rounds, where in round $t$, the player chooses $x_t \in \mathcal{X}_+$ and observes some form of feedback $\hat{f}_t$. Performance is evaluated using the expected *policy regret*,

$$\mathcal{R}_H(T) = \mathbf{E}\left[ \sum_{t=H}^{T} f_t(x_{t+1-H}, \ldots, x_t) \right] - \min_{x \in \mathcal{X}} \sum_{t=H}^{T} f_t(x, \ldots, x), \tag{3}$$

where $\mathcal{X} \subseteq \mathcal{X}_+$ is the comparator set, which may differ from the domain $\mathcal{X}_+$ where the loss functions are defined (and are well behaved). Notice that for $H = 1$, the quantity $\mathcal{R}_1(T)$ refers to the regret of standard online optimization, with no memory.

We will rely on the following conditions for the loss functions. The first is a coordinate-wise Lipschitz property, while the second is standard smoothness, stated explicitly for an $H$-coordinate setup.

**Definition 2.** $f : \mathcal{X}_+^H \to \mathbb{R}$ is coordinate-wise $L$−Lipschitz if $\forall i \in [H], x_1, \ldots, x_H, y_i \in \mathcal{X}_+$:

$$|f(x_1, \ldots, x_i, \ldots, x_H) - f(x_1, \ldots, y_i, \ldots, x_H)| \leq L\|x_i - y_i\|.$$

**Definition 3.** $f : \mathcal{X}_+^H \to \mathbb{R}$ is $\beta$−smooth if for any $x = (x_1, \ldots, x_H), y = (y_1, \ldots, y_H) \in \mathcal{X}_+^H$ :

$$f(y) - f(x) \leq \sum_{i=1}^{H} \nabla_i f(x)^\mathsf{T} (y_i - x_i) + \frac{\beta}{2} \|y_i - x_i\|^2,$$

where $\nabla_i$ is the gradient with respect to $x_i$. When $x_1 = \ldots = x_H = z$, we compress notation to $\nabla_i f(z)$.

## 2.3 Disturbance-Action Policies

Online linear control may be approximated by certain loss functions with memory, via a reparameterization suggested in [1, 2] named the Disturbance Action Policy (DAP). For completeness, we state the parameterization here even though our technical development will be mostly orthogonal.

**Definition 4** (Disturbance-Action Policy)**.** For a fixed linear controller $K_0 \in \mathcal{K}$ and parameters $M = (M^{[1]}, \ldots, M^{[H]})$ with $M^{[i]} \in \mathbb{R}^{d_u \times d_x}$, a *disturbance-action policy* chooses an action at time $t$ as

$$u_t(M) = -K_0 x_t + \sum_{i=1}^{H} M^{[i]} w_{t-i},$$

where for notational convenience we say $w_i = 0$ for $i \leq 0$.

This parameterization reduces the decision of the player at time $t$ to choosing $M_t = (M_t^{[1]}, \ldots, M_t^{[H]})$. The comparator set is given by $\mathcal{M} = \mathcal{M}^{[1]} \times \cdots \times \mathcal{M}^{[H]}$, where

$$\mathcal{M}^{[i]} = \left\{ M \in \mathbb{R}^{d_u \times d_x} \ : \ \|M\| \leq 2\kappa_B \kappa^3 (1 - \gamma)^i \right\},$$

however, the player may choose $M_t$ from the slightly larger $\mathcal{M}_+ = \{2M \ : \ M \in \mathcal{M}\}$. The following defines the adversary's cost functions, referred to as surrogate or ideal cost functions. It is a summary of Definitions 4.2, 4.4, and 4.5 of [1], as well as 3.4 of [2], and while we do not use it explicitly, we give it here for the sake of concreteness.

**Definition 5.** Denote by $M_{0:H}$ a sequence of policies $M_0, \ldots, M_H \in \mathcal{M}_+$. For a controller $K$, let $\tilde{A}_K = A_\star + KB_\star$, and define:

(1) (disturbance-state transfer matrix) $\Psi_i^K(M_{0:H-1}) = \tilde{A}_K^i \mathbb{1}_{i \leq H} + \sum_{j=1}^{H} \tilde{A}_K^j B_\star M_{H-j}^{[i-j]} \mathbb{1}_{i-j \in [1,H]}$;

(2) (ideal state) $y_{t+1}^K(M_{0:H-1}) = \sum_{i=0}^{2H} \Psi_i^K(M_{0:H-1}) w_{t-i}$;

(3) (ideal action) $v_t^K(M_{0:H}) = -K y_t^K(M_{0:H-1}) + \sum_{i=1}^{H} M_H^{[i]} w_{t-i}$.

The surrogate or ideal costs and their expected value are respectively defined as:

$$\hat{c}_t(M_{0:H}) = c_t\left(y_t^K(M_{0:H-1}), v_t^K(M_{0:H})\right), \qquad \hat{C}_t(M_{0:H}) = \mathbf{E}_w\left[c_t\left(y_t^K(M_{0:H-1}), v_t^K(M_{0:H})\right)\right].$$

The following are statements of the reduction's key results due to [2]. Denote:

$$H = \gamma^{-1} \log 2\kappa^3 T, \qquad D_{x,u} = 8\gamma^{-1} \kappa_B \kappa^3 W(H\kappa_B + 1). \tag{4}$$

The first result relates the costs $\hat{c}_t, \hat{C}_t$ and the associated regret to the original losses and regret.

**Lemma 6.** *For any algorithm $\mathcal{A}$ that plays policies $M_1, \ldots, M_T \in \mathcal{M}_+$ we have:*

(i) $\|x_t\|, \|u_t\| \leq D_{x,u}$, and thus $|c_t(x_t, u_t)| \leq C D_{x,u}^2$;

(ii) $|c_t(x_t, u_t) - \hat{c}_t(M_{t-H:t})| \leq G D_{x,u}^2 / T$;

(iii) $\overline{\mathcal{R}}_{\mathcal{A}}(T) \leq \mathbf{E}\left[\sum_{t=H+1}^{T} \hat{C}_t(M_{t-H:t}) - \min_{M_* \in \mathcal{M}} \sum_{t=H+1}^{T} \hat{C}_t(M_*, \ldots, M_*)\right] + 2D_{x,u}^2(G + HC).$

The second result establishes certain desirable properties of the cost functions $\hat{c}_t$ and $\hat{C}_t$.

**Lemma 7.** *Let $\tilde{C}_t : M \mapsto \hat{C}_t(M, \ldots, M)$. Then:*

(i) $\hat{c}_t, \hat{C}_t$ are coordinate-wise $L_f$-Lipschitz over $\mathcal{M}_+$, with $L_f = 2\kappa_B \gamma^{-1} \kappa^3 G D_{x,u} W$;

(ii) *if* $c_t(\cdot, \cdot)$ *are* $\beta$-smooth then $\hat{c}_t, \hat{C}_t, \tilde{C}_t$ are $\beta_f$-smooth over $\mathcal{M}_+$ with $\beta_f = 25\beta\kappa_B^2 \kappa^6 W^2 H / \gamma^2$;

(iii) *If* $c_t(\cdot, \cdot)$ *are* $\alpha$-strongly convex and $\mathbf{E}[w_t w_t^\mathsf{T}] \succeq \sigma^2 I$ then $\tilde{C}_t$ are $\alpha_f$−strongly convex over $\mathcal{M}$ with $\alpha_f = \frac{1}{36}\alpha\sigma^2\gamma^2/\kappa^{10}$.

We note that the second claim of Lemma 7 was not previously established. We prove it in the full version of the paper [6] using similar techniques to those used for the first claim.

---

**Algorithm 1** Bandit Linear Control

---

1: **input:** controller $K_0$, memory length $H$, step size $\eta$, and coordinate-wise sampling radii $r_t^{[i]}$
2: **Draw** $U_1 \sim \mathcal{S}^{(d_u \times d_x) \times H}$.
3: **Initialize** $\tau = 1$, $\overline{M}_1 = 0$, $M_1^{[i]} = r_1^{[i]} U_1^{[i]}$ $\quad (\forall i \in [H])$.
4: **for** $t = 1, \ldots, T$ **do**
5: $\quad$ **Play** $u_t = -K_0 x_t + \sum_{i=1}^{H} M_t^{[i]} w_{t-i}$,
6: $\quad$ **Observe** $x_{t+1}$ and $c_t(x_t, u_t)$; update $w_t = x_{t+1} - A_\star x_t - B_\star u_t$.
7: $\quad$ **Draw** $b_t \sim$ Bernoulli$(1/(2H+2))$
8: $\quad$ **if** $t \geq 2H + 2$ and $b_t \prod_{i=1}^{2H+1}(1 - b_{t-i}) = 1$ **then**
9: $\quad\quad$ **Update** $\overline{M}_{\tau+1}^{[i]} = \Pi_{\mathcal{M}^{[i]}}\big[\overline{M}_\tau^{[i]} - \eta d_x d_u H c_t(x_t, u_t) r_\tau^{[i]} U_\tau^{[i]}\big]$ $\quad (\forall i \in [H])$.
10: $\quad\quad$ $\tau \leftarrow \tau + 1$
11: $\quad\quad$ **Draw** $U_\tau \sim \mathcal{S}^{(d_u \times d_x) \times H}$.
12: $\quad\quad$ $M_{t+1}^{[i]} = \overline{M}_\tau^{[i]} + r_\tau^{[i]} U_\tau^{[i]}$ $\quad (\forall i \in [H])$.
13: $\quad$ **else**
14: $\quad\quad$ $M_{t+1} = M_t$

---

## 3 Algorithm and Main Results

We present a new algorithm for the bandit linear control problem, detailed in Algorithm 1, for which we prove:

**Theorem 8.** *Let $H, D_{x,u}, \alpha_f, \beta_f$ be as in Eq. (4) and Lemma 7, $K_0$ be a $(\kappa, \gamma)-$diagonal strongly stable controller, and $d_{min} = \min\{d_x, d_u\}$. Suppose Algorithm 1 is run with the above parameters, and*

$$\eta = \sqrt{\frac{3d_{min}^2 + (15\beta_f/\alpha_f)\log T}{Td_x^2 d_u^2 C^2 D_{x,u}^4}}, \qquad r_t^{[i]} = \big[(2\kappa_B \kappa^3 (1-\gamma)^i)^{-2} + \tfrac{1}{2}\alpha_f \eta t\big]^{-1/2}.$$

*Then the expected pseudo-regret is bounded as*

$$\overline{\mathcal{R}}_{\mathcal{A}}(T) \leq 4d_x d_u C D_{x,u}^2 (H+1)^2 \sqrt{T(3d_{min}^2 + (15\beta_f/\alpha_f)\log T)} + \tilde{O}(T^{1/4}).$$

The big-$\tilde{O}$ notation in the theorem hides polynomial dependence in the problem parameters and poly-log dependence on $T$. We prove Theorem 8 later in Section 5 after discussing our reduction technique.

There are three main components to the algorithm:

(1) A randomized schedule to determine the times of parameter updates, which ensures that these are at least $2(H+1)$ apart and $O(H)$ apart in expectation. This is part of our new reduction scheme, which is presented and discussed in Section 4.

(2) A standard one-point gradient estimate that gives a (nearly-)unbiased estimate for the gradient of a function based on bandit feedback, by perturbing the policy using uniform samples from the unit Euclidean sphere of $\mathbb{R}^{(d_u \times d_x) \times H}$; this is denoted as $U \sim \mathcal{S}^{(d_u \times d_x) \times H}$.

(3) A preconditioned (online) gradient update rule that uses mixed regularization and standard Euclidean projections $\Pi_{\mathcal{M}^{[i]}}[M] = \arg\min_{M' \in \mathcal{M}^{[i]}} \|M - M'\|_F$.

The mixed regularization, inspired by [17], is comprised of two terms (see $r_t^{[i]}$ in Theorem 8): the first exploits the strong convexity of the (expected) loss functions, while the second accounts for the small diameter of $\mathcal{M}^{[i]}$, which might be significantly smaller than the magnitude of the perturbations required for the gradient estimates (this is particularly problematic for large $i$). To avoid sampling too far away from the "admissible" set, where the cost functions are well-behaved, we cap the perturbations of the one-point estimate according to the radii $\{r_t^{[i]}\}_{i \in [H]}$ and increase the regularization term to account for the higher variance of the resulting gradient estimate.

**Computational and space complexity:** From a memory perspective, the algorithm needs to maintain the matrices $\overline{M}_\tau^{[i]}$ and $M_t^{[i]}$, which take $O(d_x d_u H)$ memory. Computationally, the bottleneck is the projection in line 9, which is onto a spectral normed sphere. This typically requires a singular

value decomposition, which takes $O(d_u^2 d_x H + d_x^3 H)$ time. However, when this is computationally prohibitive, it could be relaxed to a projection on a Frobenius normed sphere, which takes only $O(d_x d_u H)$ time, at the cost of a factor $d_{\min}$ to the regret bound in Theorem 8. Notice that the random draws in lines 2 and 11 are from the Euclidean (Frobenius normed) sphere and thus all remaining computations take $O(d_x d_u H)$ time.

## 4  Bandit Convex Optimization with Memory

In this section we give the details of our reduction from BCO with memory to standard BCO, that constitutes a key element of our main algorithm. The application to bandit linear control, however, will require a slightly more general feedback model than the usual notion of bandit feedback.

### 4.1  Setup

We consider the online optimization with memory setting described in Section 2.2, with feedback model such that on round $t$:

(1) The player chooses $x_t \in \mathcal{X}_+$, and independently, the adversary draws a random $\xi_t$;
(2) The player observes feedback $\hat{f}_t = \hat{f}_t(x_{t+1-H:t}; \xi_{t+1-H:t})$ such that, if $x_{t+1-H:t}$ are jointly independent of $\xi_{t+1-H:t}$, then $|\mathbf{E}_{\xi_{t+1-H:t}}[\hat{f}_t] - f_t(x_{t+1-H:t})| \le \varepsilon$.

The above expectation is only with respect to the variables $\xi_{t+1-H:t}$, and $\varepsilon \ge 0$ is a fixed parameter (possibly unknown to the player). Our feedback model, which may potentially seem non-standard, encompasses the following ideas, both of which are necessary for the application to linear control:

- In the standard no-memory setting ($H = 1$), standard arguments apply even if the feedback received by the learner is randomized, as long as it is independent of the learner's decision on the same round. In the memory setting, the analogous condition is that the last $H$ decisions do not depend on the adversary's randomness during this time.
- We allow feedback of the form $\hat{f}_t = f_t(x_t) + \varepsilon_t$, where $\varepsilon_t$ is a small *adaptive* adversarial disturbance that can depend on all past history (yet is at most $\varepsilon$ in absolute value).

In the context of linear control, the identity of the above terms will be $x_t := M_t$, $\xi_t := w_t$, $\hat{f}_t := c_t(x_t, u_t)$, $f_t(\cdot) := \hat{C}_t(\cdot)$, and thus $d = d_x d_u H$.

### 4.2  Base BCO Algorithm

The reduction relies on the following properties of the base algorithm $\mathcal{A}(T)$ for BCO with no memory, that can be used against an adversary that chooses loss function from $\mathcal{F} \subseteq \{f : \mathcal{X}_+ \to \mathbb{R}\}$, and observing feedback satisfying $|\mathbf{E}_{\xi_t}[\hat{f}_t] - f_t(x_t)| \le \varepsilon$ :

(i) Its regret at times $t \le T$ is bounded as $\mathcal{R}_1(t) \le R_{\mathcal{A}}(T)$ where $R_{\mathcal{A}}(T) \ge 0$;
(ii) Its predictions $x_1, \ldots, x_T$ satisfy $\|\bar{x}_{t+1} - \bar{x}_t\| \le \delta_t$ , and $\|x_t - \bar{x}_t\| \le \vartheta_t$ almost surely, where $\bar{x}_t$ is the expected value of $x_t$ conditioned on all past history up to (not including) the player's decision at time $t$, and $\delta_t, \vartheta_t$ are decreasing sequences.

The above properties are satisfied by standard BCO algorithms, often without any modification. In particular, these algorithms are amenable to our randomized and perturbed feedback model and often require only a slight modification in their regret analyses to account for the additive disturbances. (In the full version of the paper [6] we give an analysis of a concrete BCO algorithm in this setting.)

Notice that, crucially, $\delta_t$ bounds the change in the algorithm's *expected* predictions as opposed to their actual movement. This is crucial as typical BCO algorithms add large perturbations to their predictions (as part of their gradient estimation procedure), with magnitude often significantly larger than the change in the underlying expected prediction; i.e., it holds that $\vartheta_t \gg \delta_t$. Our reduction procedure is able to exploit this observation to improve performance for smooth functions.

### 4.3  The Reduction

We can now present our reduction from BCO with memory to BCO with no memory ($H = 1$); see Algorithm 2. The idea is simple: we use a base algorithm for standard BCO, denoted here by $\mathcal{A}$, using the observed feedback, but make sure that $\mathcal{A}$ is updated at most once in every $H$ rounds. Since the setup is adversarial, we cannot impose a deterministic update schedule; instead, we employ a randomized schedule in which $\mathcal{A}$ is invoked with probability $1/H$ on each round, but constrained so that this does not happen too frequently. (A similar technique was used in a different context in [12, 8].)

---

**Algorithm 2** BCO Reduction

1: **input:** memory length $H$, BCO algorithm $\mathcal{A}(T/H)$.
2: **set:** $x_1 \leftarrow \mathcal{A}.\text{initialize}()$
3: **for** $t = 1, \ldots, T$ **do**
4:     Play $x_t$ (independently, adversary draws $\xi_t$)
5:     Observe feedback $\hat{f}_t(x_{t+1-H:t}; \xi_{t+1-H:t})$
6:     Draw $b_t \sim \text{Bernoulli}(1/H)$
7:     **if** $t \geq H$ and $b_t \prod_{i=1}^{H-1}(1 - b_{t-i}) = 1$ **then**
8:         $x_{t+1} \leftarrow \mathcal{A}.\text{update}(\hat{f}_t)$
9:     **else**
10:        $x_{t+1} \leftarrow x_t$

---

The induced spacing between consecutive updates of $\mathcal{A}$ serves two purposes at once: first, it reduces the $H$-memory loss functions to functions of a single argument, amenable to optimization using $\mathcal{A}$; second, it facilitates (conditional) probabilistic independence between consecutive updates which is crucial for dealing with the extended feedback model as required by the application to linear control. (We note that these conditions are not satisfied by existing techniques [4, 3, 1].)

The following is the main result of the reduction.

**Theorem 9.** *Suppose Algorithm 2 is run using $\mathcal{A}(T/H)$ satisfying the above properties:*

   *(i) If $\tilde{f}_t : x \mapsto f_t(x, \ldots, x)$ satisfy $\tilde{f}_t \in \mathcal{F}$, and $f_t$ are coordinate-wise $L-$Lipschitz then*

$$\mathcal{R}_H(T) \leq 3HR_{\mathcal{A}}\left(\frac{T}{H}\right) + \frac{1}{2}LH^2 \sum_{t=1}^{\lfloor T/H \rfloor}(\delta_t + 2\vartheta_t);$$

   *(ii) If additionally $\tilde{f}_t$ are convex and $f_t$ are $\beta-$smooth, then*

$$\mathcal{R}_H(T) \leq 3HR_{\mathcal{A}}\left(\frac{T}{H}\right) + \frac{1}{2}H^2 \sum_{t=1}^{\lfloor T/H \rfloor+1}(L\delta_t + \beta\delta_t^2 + 6\beta\vartheta_t^2).$$

### 4.4 Proof Ideas

We provide some of the main ideas for proving Theorem 9. We start with the following technical lemma that quantifies the duration between updates of the base BCO algorithm.

**Lemma 10.** *Suppose $b_t$ in Algorithm 2 are drawn in advance for all $t \geq 1$. Let $t_0 = 0$ and for $i \geq 1$ let*

$$t_i = \min\left\{t \geq t_{i-1} + H \mid b_t \prod_{i=1}^{H-1}(1 - b_{t-i}) = 1\right\}.$$

*Then denoting $S = \{t_i \mid H \leq t_i < T\}$, the times Algorithm 2 updates $\mathcal{A}$, we have that (i) $|S| \leq \lfloor T/H \rfloor$, and (ii) $\mathbf{E}[t_i - t_{i-1}] = \mathbf{E}t_1 \leq 3H$ for all $i$.*

See proof in the full version of the paper [6]. The next lemma shows how the randomized update schedule allows us to bound the regret using the base (no memory) algorithm and an additional term that depends on the algorithm's prediction shifts.

**Lemma 11.** *Suppose Algorithm 2 is run with $\mathcal{A}(T/H)$ as in Theorem 9. If $\tilde{f}_t \in \mathcal{F}$ then we have that*

$$\mathcal{R}_H(T) \leq 3HR_{\mathcal{A}}\left(\frac{T}{H}\right) + \mathbf{E}\left[\sum_{t=H}^{T} f_t(x_{t+1-H}, \ldots, x_t) - \tilde{f}_t(x_{t+1-H})\right].$$

**Proof.** Let $S$ be the times Algorithm 2 updates $\mathcal{A}$ as defined in Lemma 10. Denote $\tilde{\xi}_t = \xi_{t+1-H:t}$, and notice that $\{\tilde{\xi}_t\}_{t \in S}$ are mutually independent since Algorithm 2 ensures there are at least $H$ rounds between updates of $\mathcal{A}$. Moreover, this implies that for any $t \in S$, $x_{t+1-H} = \ldots = x_t$, and these are also independent of $\tilde{\xi}_t$. Our $H$-memory feedback model thus implies that

$$|\tilde{f}_t(x_t) - \mathbf{E}_{\tilde{\xi}_t}[\hat{f}_t]| \leq \varepsilon, \qquad \forall t \in S,$$

and since $\tilde{f}_t \in \mathcal{F}$, we can use the regret bound of $\mathcal{A}$ to get that for any $x \in \mathcal{X}$

$$\mathbf{E}\left[\sum_{t \in S} \tilde{f}_t(x_t) - \sum_{t \in S} \tilde{f}_t(x)\right] = \mathbf{E}\left[\mathbf{E}\left[\sum_{t \in S} \tilde{f}_t(x_t) - \sum_{t \in S} \tilde{f}_t(x) \,\Big|\, S\right]\right] = \mathbf{E}[\mathcal{R}_1(|S|)] \le R_{\mathcal{A}}\left(\frac{T}{H}\right),$$

where the last transition also used the fact that $|S| \le T/H$ (see Lemma 10). Next, denote $\chi_t = b_t \prod_{i=1}^{H-1}(1 - b_{t-i})$ and notice that $\mathbf{E}\chi_t = \mathbf{E}\chi_H$. Then for any fixed $x \in \mathcal{X}$ we have that

$$\mathbf{E}\left[\sum_{t \in S} \tilde{f}_t(x)\right] = \mathbf{E}\left[\sum_{t=H}^{T} \tilde{f}_t(x)\chi_t\right] = \sum_{t=H}^{T} \tilde{f}_t(x)\mathbf{E}[\chi_t] = \mathbf{E}[\chi_H]\sum_{t=H}^{T} \tilde{f}_t(x).$$

Next, notice that $x_{t+1-H}$ is independent of $\chi_t$ and since $\chi_t = 1$ implies that $x_t = x_{t+1-H}$, we get that

$$\mathbf{E}\left[\sum_{t \in S} \tilde{f}_t(x_t)\right] = \mathbf{E}\left[\sum_{t=H}^{T} \tilde{f}_t(x_t)\chi_t\right] = \sum_{t=H}^{T} \mathbf{E}\left[\tilde{f}_t(x_{t+1-H})\right]\mathbf{E}[\chi_t] = \mathbf{E}[\chi_H]\mathbf{E}\left[\sum_{t=H}^{T} \tilde{f}_t(x_{t+1-H})\right].$$

Finally, combining the last three equations we get that

$$\mathbf{E}\left[\sum_{t=H}^{T} \tilde{f}_t(x_{t+1-H})\right] - \sum_{t=H}^{T} \tilde{f}_t(x) \le (\mathbf{E}[\chi_H])^{-1} R_{\mathcal{A}}\left(\frac{T}{H}\right) \le 3HR_{\mathcal{A}}\left(\frac{T}{H}\right),$$

where the last transition used the non-negativity of $R_{\mathcal{A}}(T/H)$ and that $\mathbf{E}[\chi_H] \ge 1/3H$. Plugging the above into $\mathcal{R}_H(T)$ concludes the proof. ∎

Completing the proof of Theorem 9 entails bounding the second term in Lemma 11. While the smooth case requires some delicate care for achieving the squared dependence on $\vartheta_t$, the proof is otherwise quite technical and thus deferred to the full version of the paper [6].

## 5  Analysis

We first require the following, mostly standard, analysis of the base procedure of Algorithm 1 for the no-memory setting ($H = 1$). See proof in the full version of the paper [6].

**Lemma 12.** *Consider the setting of Section 4 with $H = 1$ and $\varepsilon \in \tilde{O}(1/T)$, against an adversary that chooses $f_t : \mathcal{M}_+ \to \mathbb{R}$ that are $\alpha_f$-strongly convex and $\beta_f$-smooth. Let $d_{\mathcal{M}} = d_x d_u H$ be the dimension of $\mathcal{M}_+$, and $\mathcal{R}_1(t)$ be the regret of a procedure that at time $t$:*

*(i) Draws $U_t \sim \mathcal{S}^{(d_u \times d_x) \times H}$; and plays $M_t$ where $M_t^{[i]} = \overline{M}_t^{[i]} + r_t^{[i]} U_t^{[i]}$ ($\forall i \in [H]$)*
*(ii) Observes $\hat{f}_t$; and sets $\hat{g}_t^{[i]} = (d_{\mathcal{M}}/r_t^{[i]})\hat{f}_t U_t^{[i]}$ ($\forall i \in [H]$)      (1-point gradient estimate)*
*(iii) Updates $\overline{M}_{t+1}^{[i]} = \Pi_{\mathcal{M}^{[i]}}\left[\overline{M}_t^{[i]} - \eta (r_t^{[i]})^2 \hat{g}_t^{[i]}\right]$   ($\forall i \in [H]$).       (preconditioned update)*

*If $|\hat{f}_t| \le \hat{F}$, $r_t^{[i]}$, $d_{min}$ are as in Theorem 8, and $\eta \in \tilde{O}(T^{-1/2})$ then $\delta_t = d_{\mathcal{M}}\hat{F}\sqrt{2\eta/\alpha_f t}$ and $\vartheta_t^2 = 2/\alpha_f \eta t$ satisfy the assumptions of $\mathcal{A}(T)$ in Theorem 9, and for all $t \le T$*

$$\mathcal{R}_1(t) \le \frac{1}{\eta}\left(Hd_{min}^2 + \frac{2\beta_f}{\alpha_f}(1 + \log T)\right) + \frac{d_{\mathcal{M}}^2 \hat{F}^2}{2}\eta T + \tilde{O}(T^{1/4}).$$

**Proof of Theorem 8.** Consider $\hat{c}_t, \hat{C}_t$ from Definition 5 and notice that $\hat{c}_t$ depends on the last $H + 1$ policies but also the last $2(H + 1)$ system noises. This means that the effective memory of the adversary is $2(H + 1)$, prompting us to modify the definitions of $\hat{c}_t, \hat{C}_t$ to receive $2(H + 1)$ policies but ignore the first $H + 1$, i.e.,

$$\hat{C}_t^{\text{new}}(M_{0:2H+1}) = \hat{C}_t(M_{H+1:2H+1}), \qquad \hat{c}_t^{\text{new}}((M_{0:2H+1}) = \hat{c}_t(M_{H+1:2H+1}).$$

Henceforth, $\hat{c}_t, \hat{C}_t$ refer to $\hat{c}_t^{\text{new}}, \hat{C}_t^{\text{new}}$. Notice that Lemmas 6 and 7 are not impacted by this change and hold with the same $H$ as the original functions. We thus have that $\hat{C}_t$ are $L_f$-coordinate-wise Lipschitz and $\beta_f$-smooth, and $\tilde{C}_t$ are $\alpha_f$-strongly convex and $\beta_f$-smooth. Moreover, $|c_t(x_t, u_t)| \le CD_{x,u}^2$, and if $M_{t-1-2H:t}$ are independent of $w_{t-1-2H:t}$ then

$$|\mathbf{E}_{w_{t-1-2H:t}}[c_t(x_t, u_t) - \hat{c}_t(M_{t-1-2H:t})]| = |\mathbf{E}_{w_{t-1-2H:t}}[c_t(x_t, u_t)] - \hat{C}_t(M_{t-1-2H:t})| \le \frac{GD_{x,u}^2}{T}.$$

Now, Consider Algorithm 1 in the context of the BCO with memory setting presented in Section 4 with $\varepsilon = GD_{x,u}^2/T$, feedback bounded by $CD_{x,u}^2$, and let $\mathcal{R}_{2(H+1)}(T)$ be its regret against an adversary that chooses functions $f_t : \mathcal{M}_+^{2(H+1)} \to \mathbb{R}$ satisfying:

- $f_t$ are $L_f$-coordinate-wise Lipschitz and $\beta_f$-smooth;
- $\tilde{f}_t : M \mapsto f_t(M, \dots, M)$ are $\alpha_f$-strongly convex and $\beta_f$-smooth.

Since $\hat{C}_t$ satisfy these assumptions, and our choice of $r_t^{[i]}$ ensures that $M_t \in \mathcal{M}_+$, Lemma 6 yields that

$$\overline{\mathcal{R}}_{\mathcal{A}}(T) \leq \mathcal{R}_{2(H+1)}(T) + 2D_{x,u}^2(G + HC),$$

and since the second term is at most poly-log in $T$, it remains to bound $\mathcal{R}_{2(H+1)}(T)$. To that end, notice that Algorithm 1 fits the mold of our reduction procedure given in Algorithm 2 with base procedure as in Lemma 12. Now, invoking Lemma 12 with $\hat{F} = CD_{x,u}^2$ and horizon $T/2(H+1)$, the second term of Theorem 9 satisfies that

$$\sum_{t=1}^{\lfloor T/2(H+1) \rfloor + 1} (L_f \delta_t + \beta_f \delta_t^2 + 6\beta_f \vartheta_t^2) \leq \frac{12\beta_f \log T}{\alpha_f \eta} + \tilde{O}(T^{1/4}),$$

and further using Lemma 12 to bound the first term of Theorem 9, and simplifying, we get that

$$\mathcal{R}_{2(H+1)}(T) \leq 2(H+1)^2 \left[ \frac{1}{\eta}\left(3d_{\min}^2 + \frac{15\beta_f}{\alpha_f} \log T\right) + d_x^2 d_u^2 C^2 D_{x,u}^4 \eta T \right] + \tilde{O}(T^{1/4}).$$

Our choice of $\eta$ yields the final bound. ∎

# 6 Extensions to General Costs and Adversarial Noise

In this section we consider the case where the cost functions chosen by the adversary are general, possibly non-smooth (weakly) convex functions. Importantly, we also allow the system noise to be chosen by an oblivious adversary. Formally, the setup is identical to Section 2.1 but with the following modifications:

1. Only Assumptions 1-3 are assumed throughout;
2. The costs $c_t(x, u)$ are (weakly) convex functions of $(x, u)$;
3. The disturbances $w_t$ are chosen by an oblivious adversary, i.e., one that has knowledge of the algorithm but must choose all disturbances before the first round. (Notice that $w_t$ are still bounded as per Assumption 1.)

To ease notation, recall that $d_{\min} = \min\{d_x, d_u\}$, and $D^2 = \max_{M_1, M_2 \in \mathcal{M}} \|M_1 - M_2\|_F^2 \leq 4d_{\min}^2 \kappa_B^2 \kappa^6 / \gamma$. The following extends our main results, given in Theorem 8, to the setting above.

**Theorem 13.** *Let $H, D_{x,u}, L_f, \beta_f$ be as in Eq. (4) and Lemma 7, $K_0$ be a $(\kappa, \gamma)$-strongly stable controller, $d_{\mathcal{M}} = d_x d_u H$, $\hat{F} = CD_{x,u}^2$, and $r_0^{[i]} = 2\kappa_B \kappa^3 (1-\gamma)^i$. Then:*

1. *The regret of running Algorithm 1 with $r_t^{[i]} = \left[ (r_0^{[i]})^{-2} + \frac{4L_f \sqrt{(H+1)T}}{d_{\mathcal{M}} \hat{F} D} \right]^{-1/2}$, and*

$$\eta = 2 \left[ \frac{(H+1)^3 L_f^2 D^2}{d_{\mathcal{M}}^6 \hat{F}^6 T} \right]^{1/4} \text{ satisfies}$$

$$\mathcal{R}_{\mathcal{A}}(T) \leq 13\sqrt{2d_x d_u d_{min} CD_{x,u}^2 \kappa_B \kappa^3 \gamma^{-1/2} L_f (H+1)^{7/2}} T^{3/4} + \tilde{O}(T^{1/2});$$

2. *If $c_t$ are $\beta$-smooth, then the regret of running Algorithm 1 with $\eta = \left[ \frac{2(H+1)\beta_f D^2}{d_{\mathcal{M}}^4 \hat{F}^4 T} \right]^{1/3}$, and*

$$r_t^{[i]} = \left[ (r_0^{[i]})^{-2} + \left( \frac{4\beta_f^2 T}{(H+1)d_{\mathcal{M}}^2 \hat{F}^2 D^2} \right)^{1/3} \right]^{-1/2} \text{ satisfies}$$

$$\mathcal{R}_{\mathcal{A}}(T) \leq 12 \left( 2d_x d_u d_{min} CD_{x,u}^2 \kappa_B \kappa^3 \sqrt{\beta_f / \gamma} (H+1)^3 T \right)^{2/3} + \tilde{O}(T^{1/2}).$$

The proof of Theorem 13 is given in the full version of the paper [6], and follows the same ideas behind Theorem 8 with a few technical adjustments. Notice that Theorem 13 only requires a strongly stable initial controller $K_0$, as opposed to the *diagonal* strongly stable controller needed for Theorem 8. Moreover, since the noise is no longer stochastic, our definition of pseudo regret now coincides with the standard definition of regret given by $\mathcal{R}_{\mathcal{A}}(T) = \mathbf{E}\left[ \max_{K \in \mathcal{K}} R_{\mathcal{A}}(T, K) \right]$.

## Broader Impact

There are no foreseen ethical or societal consequences for the research presented herein.

## Acknowledgments and Disclosure of Funding

This work was partially supported by the Israeli Science Foundation (ISF) grant 2549/19 and by the Yandex Initiative in Machine Learning.

## Footnotes

[1]The precise $D$ for which this holds will be specified later as an explicit polynomial in the problem parameters.

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
