[Supplementary Material]

## A    Reduction to no-Memory BCO Proofs

### A.1    Proof of Theorem 9

We first need the following lemma, which bounds the prediction shifts and magnitudes of Algorithm 2.

**Lemma 14.** *Suppose Algorithm 2 is run with $\mathcal{A}(T/H)$ as in Theorem 9, and let $x_1, \ldots, x_T$ be its predictions. Then we have that for $q > 0$:*

*(i)* $\sum_{t=H}^{T} \sum_{i=2}^{H} \|x_{t+i-H} - x_{t+1-H}\| \le \sum_{t=1}^{\lfloor T/H \rfloor} (\delta_t + 2\vartheta_t);$

*(ii)* $\sum_{t=H}^{T} \sum_{i=1}^{H} \|\bar{x}_{t+i-H} - \bar{x}_{t+1-H}\|^q \le \frac{1}{2} H^2 \sum_{t=1}^{\lfloor T/H \rfloor} \delta_t^q;$

*(iii)* $\mathbf{E}\left[\sum_{t=H}^{T} \sum_{i=1}^{H} \|x_{t+i-H} - \bar{x}_{t+i-H}\|^2\right] \le 3H^2 \sum_{t=1}^{\lfloor T/H \rfloor + 1} \vartheta_t^2.$

The proof is mostly technical, and relies on the fact that Algorithm 2 changes its prediction at most once every $H$ rounds. See proof in Appendix A.2. We are now ready to prove Theorem 9.

**Proof of Theorem 9.** We show that Algorithm 2 achieves the desired regret bound. Given Lemma 11, the proof is concluded by upper bounding $\mathbf{E}\left[\sum_{t=H}^{T} f_t(x_{t+1-H}, \ldots, x_t) - \tilde{f}_t(x_{t+1-H})\right]$ under each set of assumptions. First, using the coordinate-wise Lipschitz property we get that

$$\sum_{t=H}^{T} f_t(x_{t+1-H}, \ldots, x_t) - \tilde{f}_t(x_{t+1-H}) \le L \sum_{t=H}^{T} \sum_{i=2}^{H} \|x_{t+i-H} - x_{t+1-H}\| \le \frac{1}{2} LH^2 \sum_{t=1}^{\lfloor T/H \rfloor} (\delta_t + 2\vartheta_t),$$

where the last transition follows by Lemma 14. Taking expectation concludes the first part of the proof. Now, notice that by its definition, $\bar{x}_t$ is determined given any history up to (not including) the player's decision at time $s \ge t$. Using total expectation we thus get that $\mathbf{E}[\nabla_i f_t(\bar{x}_t)^\mathsf{T} x_s] = \mathbf{E}[\nabla_i f_t(\bar{x}_t)^\mathsf{T} \bar{x}_s]$, and using this equality we get that for all $i \ge 1$,

$$\mathbf{E}\left[\nabla_i f_t(\bar{x}_{t+1-H})^\mathsf{T}(x_{t+i-H} - \bar{x}_{t+1-H})\right] = \mathbf{E}\left[\nabla_i f_t(\bar{x}_{t+1-H})^\mathsf{T}(\bar{x}_{t+i-H} - \bar{x}_{t+1-H})\right]$$
$$\le \mathbf{E}[\|\nabla_i f_t(\bar{x}_{t+1-H})\| \|\bar{x}_{t+i-H} - \bar{x}_{t+1-H}\|] \quad \text{(Cauchy-Schwarz)}$$
$$\le L\,\mathbf{E}[\|\bar{x}_{t+i-H} - \bar{x}_{t+1-H}\|], \quad (f_t \text{ Lipschitz})$$

where the last transition used the Lipschitz assumption to bound the gradient. Finally, we get that

$$\mathbf{E}\left[\sum_{t=H}^{T} f_t(x_{t+1-H}, \ldots, x_t) - \tilde{f}_t(x_{t+1-H})\right]$$

$$\le \mathbf{E}\left[\sum_{t=H}^{T} f_t(x_{t+1-H}, \ldots, x_t) - \tilde{f}_t(\bar{x}_{t+1-H})\right] \quad (\tilde{f}_t \text{ convex})$$

$$\le \mathbf{E}\left[\sum_{t=H}^{T} \sum_{i=1}^{H} \nabla_i f_t(\bar{x}_{t+1-H})^\mathsf{T}(x_{t+i-H} - \bar{x}_{t+1-H}) + \frac{\beta}{2}\|x_{t+i-H} - \bar{x}_{t+1-H}\|^2\right] \quad (f_t \text{ smooth})$$

$$\le \mathbf{E}\left[\sum_{t=H}^{T} \sum_{i=1}^{H} L\|\bar{x}_{t+i-H} - \bar{x}_{t+1-H}\| + \beta\|\bar{x}_{t+i-H} - \bar{x}_{t+1-H}\|^2 + \beta\|x_{t+i-H} - \bar{x}_{t+i-H}\|^2\right],$$

where the last transition used the previous equation and the triangle inequality. Plugging in the expressions provided in Lemma 14 concludes the proof. ∎

### A.2    Proofs of Lemmas 10 and 14

**Proof of Lemma 10.** First, notice that by definition $t_i - t_{i-1} \ge H$. Summing over $i$ and recalling $t_0 = 0$ we get that $t_i \ge iH$. By definition of $S$ we then get that $|S|H \le t_{|S|} < T$, and changing sides concludes the first part of the lemma.

To see the second part of the lemma, consider a Markov chain with states corresponding to $H$-tuples of bits that captures the evolution of the sequence $(b_{t-H+1}, \ldots, b_t)$ as $t$ increases. Notice that our quantity of interest is the expected return time of the state $s = (0, 0, \ldots, 1)$. Since the chain is irreducible it admits a stationary distribution $\pi^*$, and by a standard fact about Markov chains (e.g., Proposition 1.14 in [21]), the desired expected return time equals $1/\pi^*(s)$. The latter probability equals $(1/H)(1 - 1/H)^{H-1} \ge 1/(eH) \ge 1/(3H)$, which gives the claim. ∎

**Proof of Lemma 14.** For the first claim, noticing that the algorithm only changes predictions at times $t \in S$, we have that

$$\sum_{t=H}^{T} \sum_{i=2}^{H} \|x_{t+i-H} - x_{t+1-H}\| \leq \sum_{i=2}^{H} \sum_{j=1}^{i-1} \sum_{t=H}^{T} \|x_{t+j+1-H} - x_{t+j-H}\| \qquad \text{(triangle in.Eq)}$$

$$\leq \frac{1}{2} H^2 \sum_{t=1}^{T-1} \|x_{t+1} - x_t\|$$

$$= \frac{1}{2} H^2 \sum_{t \in S} \|x_{t+1} - x_t\|$$

$$\leq \frac{1}{2} H^2 \sum_{t \in S} \|x_{t+1} - \bar{x}_{t+1}\| + \|\bar{x}_{t+1} - \bar{x}_t\| + \|\bar{x}_t - x_t\| \qquad \text{(triangle in.Eq)}$$

$$\leq \frac{1}{2} H^2 \sum_{t=1}^{|S|} \vartheta_{t+1} + \delta_t + \vartheta_t$$

$$\leq \frac{1}{2} H^2 \sum_{t=1}^{\lfloor T/H \rfloor} \delta_t + 2\vartheta_t,$$

where the last transition also used the decreasing property of $\vartheta_t$. Next, we have that for any $q > 0$

$$\sum_{t=H}^{T} \sum_{i=1}^{H} \|\bar{x}_{t+i-H} - \bar{x}_{t+1-H}\|^q = \sum_{t=H}^{T} \sum_{i=2}^{H} \| \sum_{j=1}^{i-1} \bar{x}_{t+j+1-H} - \bar{x}_{t+j-H}\|^q$$

$$= \sum_{i=2}^{H} \sum_{j=1}^{i-1} \sum_{t=H}^{T} \|\bar{x}_{t+j+1-H} - \bar{x}_{t+j-H}\|^q$$

$$\leq \frac{1}{2} H^2 \sum_{t=1}^{T-1} \|\bar{x}_{t+1} - \bar{x}_t\|^q$$

$$\leq \frac{1}{2} H^2 \sum_{t=1}^{|S|} \delta_t^q$$

$$\leq \frac{1}{2} H^2 \sum_{t=1}^{\lfloor T/H \rfloor} \delta_t^q,$$

where the second transition follows since predictions change at most once every $H$ rounds and thus there is at most one summand that is non-zero. This concludes the second part of the lemma. Next, recall that $t_i$ from Lemma 10 are the times Algorithm 2 updates the base BCO $\mathcal{A}$, and subsequently its prediction. Then we get that

$$\mathbf{E}\left[ \sum_{t=H}^{T} \sum_{i=1}^{H} \|x_{t+i-H} - \bar{x}_{t+i-H}\|^2 \right] \leq H\mathbf{E}\left[ \sum_{t=1}^{T} \|x_t - \bar{x}_t\|^2 \right]$$

$$\leq H\mathbf{E}\left[ \sum_{s=1}^{|S|+1} \vartheta_s^2 (t_s - t_{s-1}) \right]$$

$$\leq H \sum_{s=1}^{\lfloor T/H \rfloor + 1} \vartheta_s^2 \mathbf{E}[t_s - t_{s-1}]$$

$$\leq 3H^2 \sum_{t=1}^{\lfloor T/H \rfloor + 1} \vartheta_t^2.$$

where the last two transitions used Lemma 10. ∎

# B  Base BCO Algorithm

We give a general example of a BCO algorithm that may be employed in conjunction with our reduction procedure given in Algorithm 2. For a positive semi-definite matrix $P \in \mathbb{R}^{d \times d}$ define the projection in $\|\cdot\|_P$ distance $\Pi_{\mathcal{X}}^P(x) = \arg\min_{y \in \mathcal{X}} \|x - y\|_P$, where $\|x\|_P^2 = x^\mathsf{T} P x$. We analyze Algorithm 3, a standard BCO procedure that uses a preconditioned gradient update, and a one-point gradient estimate.

---

**Algorithm 3** Base BCO

1: **input:** regularization matrices $P_t \geq 0$, step size $\eta$
2: **set:** $\bar{x}_1 \in \mathcal{X}$
3: **for** $t = 1, \dots, T$ **do**
4:      Draw $u_t \sim \mathbb{S}^d$
5:      Play $x_t = \bar{x}_t + P_t^{-1/2} u_t$
6:      Observe $\hat{f}_t$ and set $\hat{g}_t = d \hat{f}_t P_t^{1/2} u_t$
7:      Update $\bar{x}_{t+1} = \Pi_{\mathcal{X}}^{P_t} (\bar{x}_t - \eta P_t^{-1} \hat{g}_t)$

---

Since our setting of BCO with (no) memory $H = 1$ uses a non-standard feedback model, we provide a full analysis of the bounds on the regret, and the prediction shifts and magnitudes. To that end, denote

$$D = \max_{x,y \in \mathcal{X}} \|x - y\|, \qquad D_P = \max_{x,y \in \mathcal{X}} \|x - y\|_P, \qquad \hat{F} = \max_{t \in [T]} |\hat{f}_t|.$$

**Lemma 15.** *Consider the BCO with no memory ($H = 1$) setting described in Section 4.2 against an adversary that chooses $f_t : \mathcal{X}_+ \to \mathbb{R}$ that are $\alpha-$strongly convex over $\mathcal{X}$ ($\alpha = 0$ in the weakly convex case). If Algorithm 3 is run with regularization matrices $P_t = P_0 + \frac{1}{2} \alpha \eta t I$ where $P_0 \geq 0$, then*

$$\delta_t = d \eta \hat{F} \|P_t^{-1/2}\| \qquad \vartheta_t^2 = \|P_t^{-1}\|$$

*satisfy the assumptions of $\mathcal{A}(T)$ in Theorem 9. Moreover, for all $t \leq T$ we have that*

1. *if $f_s$ are L-Lipschitz then $\mathcal{R}_1(t) \leq \frac{D_{P_1}^2}{\eta} + \frac{\eta d^2 \hat{F}^2}{2} t + d \varepsilon t D_{P_t} + 2L \sum_{s=1}^{t} \|P_s^{-1/2}\|$;*

2. *if $f_s$ are $\beta$-smooth then $\mathcal{R}_1(t) \leq \frac{D_{P_1}^2}{\eta} + \frac{\eta d^2 \hat{F}^2}{2} t + d \varepsilon t D_{P_t} + \beta \sum_{s=1}^{t} \|P_s^{-1}\|$.*

The proof of Lemma 15 relies on a few standard results. First, we require a standard regret bound for the time-varying preconditioned update rule. This is stated in the next lemma, which is is a specialization of bounds found in, e.g., [14], to the case of strongly convex quadratic regularizers.

**Lemma 16.** *Let $\hat{g}_1, \dots, \hat{g}_t \in \mathbb{R}^d$, and $P_t \geq \dots \geq P_1 > 0$ be arbitrary. For step size $\eta > 0$ define the update rule: $\bar{x}_{t+1} = \Pi_{\mathcal{X}}^{P_t} (\bar{x}_t - \eta P_t^{-1} \hat{g}_t)$. Then we have that*

$$\sum_{s=1}^{t} \hat{g}_s^\mathsf{T} (\bar{x}_s - x^*) \leq \frac{1}{\eta} \|\bar{x}_1 - x^*\|_{P_1}^2 + \frac{1}{\eta} \sum_{s=2}^{t} \|\bar{x}_s - x^*\|_{P_s - P_{s-1}}^2 + \frac{\eta}{2} \sum_{s=1}^{t} \|\hat{g}_s\|_{P_s^{-1}}^2, \quad \forall x^* \in \mathcal{X}.$$

Next, we need the notion of smoothing and the one point-gradient estimate, which were initially proposed by [15] and later refined in [25, 17]. The following lemma due to [17] encapsulates the relevant results.

**Lemma 17** (Lemmas 6 and 7 in [17]). *Let $P \in \mathbb{R}^{d \times d}$ be symmetric and non-singular, $b \sim \mathcal{B}^d$, and $u \sim \mathbb{S}^d$. Define the smoothed version of $f : \mathcal{X}_+ \to \mathbb{R}$ with respect to $P$ as*

$$\bar{f}(x) = \mathbf{E}[f(x + Pb)].$$

*Then we have that:*

   *(i) $\nabla \bar{f}(x) = \mathbf{E}[df(x + Pu) P^{-1} u]$;*
   *(ii) if $f$ is $\alpha-$strongly convex then so is $\bar{f}$;*
   *(iii) if $f$ is convex and $\beta-$smooth then $0 \leq \bar{f}(x) - f(x) \leq \frac{\beta}{2} \|P^2\|$, $\forall x \in \mathcal{X}_+$;*
   *(iv) if $f$ is convex and L-Lipschitz then $0 \leq \bar{f}(x) - f(x) \leq L\|P\|$, $\forall x \in \mathcal{X}_+$.*

Among other things, this lemma implies that a regret bound for a sequence $\bar{f}_t$ yields one for $f_t$. We are now ready to prove Lemma 15.

**Proof of Lemma 15.** First notice that $\bar{x}_t$ in Algorithm 3 is indeed the expectation of $x_t$ conditioned on all past history up to (not including) the decision at time $t$ (since $u_t$ is a zero mean independent random variable). Using the projection's shrinking property we get that

$$
\begin{aligned}
\|\bar{x}_{t+1} - \bar{x}_t\| &\le \|P_t^{-1/2}\| \|\bar{x}_{t+1} - \bar{x}_t\|_{P_t} \\
&\le \|P_t^{-1/2}\| \|\eta d \hat{f}_t P_t^{-1/2} u_t\|_{P_t} \\
&= \|P_t^{-1/2}\| \eta d |\hat{f}_t| \\
&\le d\eta \hat{F} \|P_t^{-1/2}\| = \delta_t.
\end{aligned}
$$

Next, we have that $\|x_t - \bar{x}_t\|^2 = \|P_t^{-1/2} u_t\|^2 \le \|P_t^{-1}\| = \vartheta_t^2$, thus concluding first part of the proof. Moving on to the regret bound, let $x \in \mathcal{X}$ be fixed, and denote $g_s = df_s(\bar{x}_s) P_s^{1/2} u_s$, the desired gradient estimate at time $s$. Recalling that $\bar{x}_s$ is independent of the adversary's random variable $\xi_s$, we use total expectation to get that

$$
\begin{aligned}
\mathbf{E}\left[\sum_{s=1}^{t}(g_s - \hat{g}_s)^\top(\bar{x}_s - x)\right] &= d\mathbf{E}\left[\sum_{s=1}^{t}(f_s(\bar{x}_s) - \hat{f}_s) u_s^\top P_s^{1/2}(\bar{x}_s - x)\right] \\
&= d\mathbf{E}\left[\sum_{s=1}^{t}(f_s(\bar{x}_s) - \mathbf{E}_{\xi_s}[\hat{f}_s]) u_s^\top P_s^{1/2}(\bar{x}_s - x)\right] \\
&\le d\varepsilon \mathbf{E}\left[\sum_{s=1}^{t}\|\bar{x}_s - x\|_{P_s}\right] \le d\varepsilon t D_{P_t},
\end{aligned}
$$

where the second to last transition used the Cauchy-Schwarz inequality, and the last transition used the assumption that $P_s$ is increasing. Next, notice that $\bar{x}_s, P_s, \hat{g}_s$ satisfy the conditions of Lemma 16, and since $\|\hat{g}_s\|_{P_s^{-1}}^2 \le d^2 \hat{F}^2$, we get that

$$
\sum_{s=1}^{t} \hat{g}_s^\top(\bar{x}_s - x) \le \frac{D_{P_1}^2}{\eta} + \frac{\alpha}{2}\sum_{s=1}^{t}\|\bar{x}_s - x\|^2 + \frac{\eta d^2 \hat{F}^2}{2} t,
$$

and taking expectation, summing the last two equations, and changing sides, we get that

$$
\mathbf{E}\left[\sum_{s=1}^{t}\bar{f}_s(\bar{x}_s) - \sum_{s=1}^{t}\bar{f}_s(x)\right] \le \mathbf{E}\left[\sum_{s=1}^{t}g_s^\top(\bar{x}_s - x) - \frac{\alpha}{2}\sum_{s=1}^{t}\|\bar{x}_s - x\|^2\right] \le \frac{D_{P_1}^2}{\eta} + \frac{\eta d^2 \hat{F}^2}{2} t + d\varepsilon t D_{P_t},
$$
(5)

where the first transition also used Lemma 17 to show that $g_s$ is an unbiased estimate of $\nabla \bar{f}_s(\bar{x}_s)$ given $\bar{x}_s$, and that $\bar{f}_s$ are $\alpha$ strongly convex (with $\alpha = 0$ in the weakly convex case). Now, let $\bar{f}_s$ be smoothed with respect to $P_s^{-1/2}$ as defined in Lemma 17. If $f_s$ are $\beta$ smooth, we get that

$$
\begin{aligned}
\mathbf{E}[f_s(x_s)] &\le \mathbf{E}\left[f_s(\bar{x}_s) + \nabla f_s(\bar{x}_s)^\top P_s^{-1/2} u_s + \frac{\beta}{2}\|P_s^{-1}\|\right] \\
&= \mathbf{E}[f_s(\bar{x}_s)] + \frac{\beta}{2}\|P_s^{-1}\| \\
&\le \mathbf{E}[\bar{f}_s(\bar{x}_s)] + \frac{\beta}{2}\|P_s^{-1}\|,
\end{aligned}
$$

where the last transition used Lemma 17, which also gives us that $-f_s(x) \le -\bar{f}_s(x) + \frac{\beta}{2}\|P_s^{-1}\|$. We thus conclude that

$$
\mathcal{R}_1(t) = \max_{x \in \mathcal{X}}\left\{\mathbf{E}\left[\sum_{s=1}^{t}f_s(x_s) - \sum_{s=1}^{t}f_s(x)\right]\right\} \le \max_{x \in \mathcal{X}}\left\{\mathbf{E}\left[\sum_{s=1}^{t}\bar{f}_s(\bar{x}_s) - \sum_{s=1}^{t}\bar{f}_s(x)\right]\right\} + \beta\sum_{s=1}^{t}\|P_s^{-1}\|,
$$

and plugging in Eq. (5) concludes the smooth case. Finally, if $f_s$ are $L$ Lipschitz then using Lemma 17 we get that

$$
\mathbf{E}[f_s(x_s) - f_s(x)] \le \mathbf{E}[f_s(\bar{x}_s) - f_s(x)] + L\|P_s^{-1/2}\| \le \mathbf{E}[\bar{f}_s(\bar{x}_s) - \bar{f}_s(x)] + 2L\|P_s^{-1/2}\|,
$$

and summing over $s$ and plugging Eq. (5) concludes the non-smooth case. ∎

## C Main Result Proofs

### C.1 Proof of Lemma 12

This lemma is a direct specification of Lemma 15 with the appropriate choice of parameters. For $M \in \mathcal{M}_+$, denote $[M]_{\text{vec}} \in \mathbb{R}^{d_x d_u H}$ the column stacking of $M$. Next, denote $I \in \mathbb{R}^{d_x d_u \times d_x d_u}$, the identity matrix, and $\text{diag}(r_t^{[1]}, \ldots, r_t^{[H]}) \in \mathbb{R}^{H \times H}$, the diagonal matrix with $r_t \in \mathbb{R}^H$ on its diagonal. Consider Algorithm 3 with $\mathcal{X} = \mathcal{M}$ (column stacked), dimension $d_{\mathcal{M}} = d_x d_u H$, and

$$P_t = [\text{diag}(r_t^{[1]}, \ldots, r_t^{[H]}) \otimes I]^{-2},$$

where $\otimes$ is the Kronecker product. Then, first, for $M \in \mathcal{M}_+$ we can write the projection as

$$\Pi_{\mathcal{M}}^{P_t}([M]_{\text{vec}}) = \underset{M' \in \mathcal{M}}{\arg\min} \|[M]_{\text{vec}} - [M']_{\text{vec}}\|_{P_t}^2 = \underset{M' \in \mathcal{M}}{\arg\min} \sum_{i=1}^{H} (r_t^{[i]})^{-2} \|M^{[i]} - M'^{[i]}\|_F^2,$$

and since $\mathcal{M} = \mathcal{M}^{[1]} \times \ldots \times \mathcal{M}^{[H]}$, each term in the sum may be minimized separately, and so we get that

$$\Pi_{\mathcal{M}}^{P_t}([M]_{\text{vec}}) = [\Pi_{\mathcal{M}^{[1]}}(M^{[1]}), \ldots, \Pi_{\mathcal{M}^{[H]}}(M^{[H]})]_{\text{vec}},$$

where $\Pi_{\mathcal{M}^{[i]}}(M) = \arg\min_{M' \in \mathcal{M}^{[i]}} \|M - M'\|$. Second, we have that $\hat{g}_t = [\hat{g}_t^{[1]}, \ldots, \hat{g}_t^{[H]}]_{\text{vec}}$ and thus the update rule may be rewritten as

$$[\overline{M}_{t+1}]_{\text{vec}} = [\overline{M}_t]_{\text{vec}} - \eta P_t^{-1} \hat{g}_t = [\overline{M}_t^{[1]} - \eta (r_t^{[1]})^2 \hat{g}_t^{[1]}, \ldots, \overline{M}_t^{[H]} - \eta (r_t^{[H]})^2 \hat{g}_t^{[H]}]_{\text{vec}}.$$

We conclude that the procedure in Lemma 12 is indeed described by Algorithm 3. We can now conclude the lemma using Lemma 15. A simple calculation shows that

$$D^2 = \max_{M_1, M_2 \in \mathcal{M}} \|M_1 - M_2\|_F^2 \leq 4 d_{\min}^2 \kappa_B^2 \kappa^6 / \gamma;$$

$$D_{P_0}^2 = \max_{M_1, M_2 \in \mathcal{M}} \|M_1 - M_2\|_{P_0}^2 \leq H d_{\min}^2.$$

Moreover, $D_{P_t}^2 = D_{P_0}^2 + \frac{1}{2} \alpha \eta t D^2$, and $\|P_t^{-1}\| \leq \frac{2}{\alpha \eta t}$. Plugging this into Lemma 15 we get that for all $t \leq T$

$$\mathcal{R}_1(t) \leq \frac{1}{\eta} \left( H d_{\min}^2 + \frac{2\beta_f}{\alpha_f} (1 + \log T) \right) + \frac{d_{\mathcal{M}}^2 \hat{F}^2}{2} \eta T$$

$$+ \underbrace{\frac{2 d_{\min}^2 \kappa_B^2 \kappa^6 \alpha_f}{\gamma} + d_{\mathcal{M}} \varepsilon \left( \sqrt{H} d_{\min} T + \sqrt{\frac{4 \alpha_f \eta d_{\min}^2 \kappa_B^2 \kappa^6 T^3}{\gamma}} \right)}_{R_{\text{low}}},$$

and for $\varepsilon \in \tilde{O}(T^{-1})$, and $\eta \in \tilde{O}(T^{-1/2})$, we indeed have that $R_{\text{low}} \in \tilde{O}(T^{-1/4})$. Finally, $\delta_t, \vartheta_t$ translate directly between lemmas, thus concluding the proof.

### C.2 Low Order Terms in Theorem 8

We summarize the low order terms that were omitted in the last three equations of the proof of Theorem 8 given in Section 5. The first of the three explicitly states the lower order term

$$R_{\text{low}}^{(1)} = 2 D_{x,u}^2 (G + HC),$$

which is later omitted in the last step. The second equation results from invoking Lemma 12 with $\hat{F} = C D_{x,u}^2$ and horizon $T/2(H+1)$, to bound the second term of Theorem 9. Here the terms related to $\delta_t, \delta_t^2$ were omitted, and satisfy

$$\sum_{t=1}^{\lfloor T/2(H+1) \rfloor + 1} (L_f \delta_t + \beta_f \delta_t^2) \leq 2 L_f d_x d_u C D_{x,u}^2 \sqrt{\frac{\eta H T}{\alpha_f}} + \frac{2 \beta_f d_x^2 d_u^2 H^2 C^2 D_{x,u}^4 \eta \log T}{\alpha_f} = R_{\text{low}}^{(2)}.$$

The third equation results from plugging in the previous result as well as that of Lemma 12 with horizon $T/(2(H+1))$, and $\varepsilon = G D_{x,u}^2 / T$ into Theorem 9. Lemma 12 yields a low order term, which

is given at the end of the proof in Appendix C.1. Plugging in the horizon, $\varepsilon$, and $\hat{F}$ this term is given by

$$R_{\text{low}}^{(3)} = \frac{2d_{\min}^2 \kappa_B^2 \kappa^6 \alpha_f}{\gamma} + d_x^2 d_u^2 H^2 G D_{x,u}^2 \left( \sqrt{H} d_{\min} + \sqrt{\frac{2\alpha_f \eta d_{\min}^2 \kappa_B^2 \kappa^6 T}{\gamma(H+1)}} \right),$$

and thus the final low order term is given by

$$R_{\text{low}} = R_{\text{low}}^{(1)} + 2(H+1)^2 R_{\text{low}}^{(2)} + 6(H+1) R_{\text{low}}^{(3)}.$$

Since $H$ is logarithmic in $T$, and $\eta \in \tilde{O}(T^{-1/2})$, we get that $R_{\text{low}} \in \tilde{O}(T^{1/4})$, as desired.

## C.3 Proof of (ii) in Lemma 7

**Proof.** Recall from Definition 5 that $\hat{c}_t(M_{0:H}) = c_t(y_t(M_{0:H}), v_t(M_{0:H}))$, and denote

$$z_t(M_{0:H}) = [y_t(M_{0:H})^\mathsf{T} v_t(M_{0:H})^\mathsf{T}]^\mathsf{T}.$$

Since $z_t(\cdot)$ is a linear mapping, its Jacobian is constant, and we denote it as $J_{z_t}$. Applying the chain rule, we get that for all $M_0, \dots, M_H \in \mathcal{M}_+$

$$\|\nabla^2 \hat{c}_t(M_{0:H})\| = \|J_{z_t}^\mathsf{T} \nabla^2 c_t(y_t(M_{0:H}), v_t(M_{0:H})) J_z\| \le \beta \|J_{z_t}\|^2,$$

and thus bounding $\|J_{z_t}\|$ will show that $\hat{c}_t$ is smooth. To that end, notice that an intermediate step of Lemma 5.6 of [1] shows that for any $M_0, \dots, M_H, M'_0, \dots, M'_H \in \mathcal{M}_+$, we have that

$$\|z_t(M_{0:H}) - z_t(M_0, \dots, M'_{H-k}, \dots, M_H)\| \le 5\kappa_B \kappa^3 W (1-\gamma)^k \sum_{i=0}^H \|M_{H-k}^{[i]} - M'^{[i]}_{H-k}\|. \quad (6)$$

Recalling that $\|\cdot\| \le \|\cdot\|_F$, and using the triangle and Cauchy-Schwarz inequalities we get that

$$\|z_t(M_{0:H}) - z_t(M'_{0:H})\| \le 5\kappa_B \kappa^3 W \sum_{k=0}^H (1-\gamma)^k \sum_{i=1}^H \|M_{H-k}^{[i]} - M'^{[i]}_{H-k}\|_F$$

$$\le 5\kappa_B \kappa^3 W \sqrt{H} \sum_{k=0}^H (1-\gamma)^k \|M_{H-k} - M'_{H-k}\|_F$$

$$\le 5\kappa_B \kappa^3 W \sqrt{H} \|M_{0:H} - M'_{0:H}\|_F \sqrt{\sum_{k=0}^H (1-\gamma)^{2k}}$$

$$\le 5\kappa_B \kappa^3 W \sqrt{\frac{H}{\gamma}} \|M_{0:H} - M'_{0:H}\|_F.$$

Since $\mathcal{M}_+$ contains an open set of $\mathbb{R}^{(d_u \times d_x) \times (H+1)}$, this Lipschitz property implies that $\|J_z\| \le 5\kappa_B \kappa^3 W \sqrt{\frac{H}{\gamma}} \le \sqrt{\beta_f/\beta}$, thus showing that $\hat{c}_t$ is $\beta_f$ smooth. Since $\hat{C}_t$ results from taking expectation of $\hat{c}_t$ with respect to the random system noise, it is also $\beta_f$ smooth.

Next, recall that $\tilde{c}_t(M) = \hat{c}_t(M, \dots, M)$, and thus defining $\tilde{z}_t = z_t(M, \dots, M)$, and repeating the process above, it suffices to show that $\tilde{z}_t$ is $\sqrt{\beta_f/\beta}$ Lipschitz to conclude that $\tilde{c}_t, \tilde{C}_t$ are $\beta_f$ smooth. Using Eq. (6) we get that for $M, M' \in \mathcal{M}_+$

$$\|\tilde{z}_t(M) - \tilde{z}_t(M')\| \le 5\kappa_B \kappa^3 W \sum_{k=0}^H (1-\gamma)^k \sum_{i=1}^H \|M^{[i]} - M'^{[i]}\|_F$$

$$\le 5\kappa_B \kappa^3 W \sqrt{H} \|M - M'\|_F \sum_{k=0}^H (1-\gamma)^k$$

$$\le \frac{5\kappa_B \kappa^3 W}{\gamma} \sqrt{H} \|M - M'\|_F$$

$$= \sqrt{\beta_f/\beta} \|M - M'\|_F,$$

thus establishing the Lipschitz property and concluding the proof. ∎

# D  Extensions Proofs

We first need to extend the base BCO procedure to the weakly convex cases. Similarly to Lemma 12, this is an immediate corollary of Lemma 15 with appropriate choice of parameters.

**Lemma 18.** *Consider the setting of Section 4 with $H = 1$ and $\varepsilon \in \tilde{O}(1/T)$, against an adversary that chooses $f_t : \mathcal{M}_+ \to \mathbb{R}$ that are convex. Let $d_\mathcal{M}, D, r_0^{[i]}$ be as in Theorem 13, and $\mathcal{R}_1(t)$ be the regret of a procedure that at time $t$:*

   *(i)* *Draws $U_t \sim \mathcal{S}^{(d_u \times d_x) \times H}$; and plays $M_t$ where $M_t^{[i]} = \overline{M}_t^{[i]} + r_t^{[i]} U_t^{[i]}$ ($\forall i \in [H]$)*
   *(ii)* *Observes $\hat{f}_t$; and sets $\hat{g}_t^{[i]} = (d_\mathcal{M}/r_t^{[i]}) \hat{f}_t U_t^{[i]}$ ($\forall i \in [H]$)        (1-point gradient estimate)*
   *(iii)* *Updates $\overline{M}_{t+1}^{[i]} = \Pi_{\mathcal{M}^{[i]}} [\overline{M}_t^{[i]} - \eta (r_t^{[i]})^2 \hat{g}_t^{[i]}]$    ($\forall i \in [H]$).        (preconditioned update)*

*Suppose that $|\hat{f}_t| \le \hat{F}$ then for all $t \le T$:*

   *1. if $f_t$ are $L$-Lipschitz, $\eta = 2 \left[ \frac{L^2 D^2}{d_\mathcal{M}^6 D^6 T} \right]^{1/4}$, and $r_t^{[i]} = \left[ (r_0^{[i]})^{-2} + \frac{4L\sqrt{T}}{d_\mathcal{M} \hat{F} D} \right]^{-1/2}$ then*

$$\mathcal{R}_1(t) \le 4\sqrt{d_\mathcal{M} L D \hat{F}} T^{3/4} + \tilde{O}(T^{1/4}), \qquad \delta_t = \frac{D}{\sqrt{T}}, \qquad \vartheta_t^2 = \frac{d_\mathcal{M} D \hat{F}}{4L\sqrt{T}};$$

   *2. if $f_t$ are $\beta_f$ smooth, $\eta = \left[ \frac{2\beta_f D^2}{d_\mathcal{M}^4 \hat{F}^4 T} \right]^{1/3}$, and $r_t^{[i]} = \left[ (r_0^{[i]})^{-2} + (\frac{4\beta_f^2 T}{d_\mathcal{M}^2 \hat{F}^2 D^2})^{1/3} \right]^{-1/2}$ then*

$$\mathcal{R}_1(t) \le (4\sqrt{\beta_f} d_\mathcal{M} \hat{F} D T)^{2/3} + \tilde{O}(T^{1/3}), \qquad \delta_t = \frac{D}{\sqrt{T}} \qquad \vartheta_t^2 = \left( \frac{d_\mathcal{M}^2 \hat{F}^2 D^2}{4\beta_f^2 T} \right)^{1/3}.$$

See proof in Appendix D.1.

**Proof of Theorem 13.** First, unlike the proof of Theorem 8, here we use $\hat{c}_t$ as given in Definition 5, without any modification. As before we view Algorithm 1 in the context of the BCO with memory setting presented in Section 4. The adversary's noise $\xi_t$ is now degenerate ($w_t$ are not stochastic), the costs are given by $\hat{c}_t$ and by Lemma 6, the feedback satisfies

$$|c_t(x_t, u_t) - \hat{c}_t(M_{t-1-2H:t})| \le \frac{GD_{x,u}^2}{T},$$

and thus $\varepsilon = GD_{x,u}^2/T$, and the feedback $c_t(x_t, u_t)$ is bounded by $CD_{x,u}^2$. Let $\mathcal{R}_{H+1}(T)$ be the regret of Algorithm 1 against an adversary with memory $H + 1$, and notice that our choice of $r_t^{[i]}$, and in particular $r_0^{[i]}$, ensures that $M_t \in \mathcal{M}_+$. Since the third part of Lemma 6 is, in fact, proven for $\hat{c}_t$ rather than $\hat{C}_t$ (the latter is an immediate corollary) and so we have that

$$\mathcal{R}_\mathcal{A}(T) \le \mathcal{R}_{H+1}(T) + 2D_{x,u}^2(G + HC).$$

Since the second term is at most poly-log in $T$, it remains to bound $\mathcal{R}_{H+1}(T)$ for each set of assumptions and parameter choices. Recall that Algorithm 1 fits the mold of our reduction procedure given in Algorithm 2 with base procedure as in Lemma 18. Moving forward, our analysis is divided in two. Consider the first set of parameter choices (with no smoothness assumptions). By Lemma 7, $\hat{c}_t$ are coordinate-wise $L_f$ Lipschitz, and thus $\tilde{c}_t : M \mapsto \hat{c}_t(M, \ldots, M)$ are $(H + 1)L_f$ Lipschitz. Invoking Lemma 18 with $\hat{F} = CD_{x,u}^2$, $L = (H + 1)L_f$ and horizon $T/(H + 1)$, the second term of Theorem 9 (with no smoothness assumption) satisfies that

$$\frac{1}{2} L_f (H + 1)^2 \sum_{t=1}^{\lfloor T/(H+1) \rfloor} \delta_t + 2\vartheta_t \le \frac{1}{2} L_f (H + 1) T (\delta_t + 2\vartheta_t)$$

$$\le \frac{1}{2} L_f (H + 1) T \left( D\sqrt{\frac{H + 1}{T}} + \sqrt{\frac{d_\mathcal{M} \hat{F} D}{L_f}} \left( \frac{H + 1}{T} \right)^{1/4} \right)$$

$$\le \frac{1}{2} \sqrt{d_\mathcal{M} \hat{F} D L_f (H + 1)^{5/2}} T^{3/4} + \tilde{O}(T^{1/2}),$$

and further using Lemma 18 to bound the first term of Theorem 9, and simplifying, we get that

$$\mathcal{R}_{H+1}(T) \leq 13\sqrt{d_{\mathcal{M}} \hat{F} D L_f (H+1)^{5/2} T^{3/4}} + \tilde{O}(T^{1/2})$$

$$\leq 13\sqrt{2 d_x d_u d_{\min} C D_{x,u}^2 \kappa_B \kappa^3 \gamma^{-1/2} L_f (H+1)^{7/2} T^{3/4}} + \tilde{O}(T^{1/2}),$$

where the last step only plugs in the values of $d_{\mathcal{M}}, \hat{F}, D$. This concludes the proof of the non-smooth case. Now suppose that $c_t$ are $\beta$ smooth and Algorithm 1 is run with our second choice of parameters. Notice that the proof of the smoothness in Lemma 7 (see Appendix C.3) actually shows that both $\hat{c}_t, \tilde{c}_t$ are $\beta_f$ smooth, and as before, we invoke Lemma 18 with our parameter choices to bound the second term of Theorem 9 (with the smoothness assumption) by

$$\frac{1}{2}(H+1)^2 \sum_{t=1}^{\lfloor T/2(H+1) \rfloor + 1} ((H+1)L_f \delta_t + \beta_f \delta_t^2 + 6\beta_f \vartheta_t^2)$$

$$\leq (H+1)T((H+1)L_f \delta_t + \beta_f \delta_t^2 + 6\beta_f \vartheta_t^2)$$

$$\leq (H+1)T\left( L_f D \sqrt{\frac{(H+1)^3}{T}} + \beta_f D^2 \frac{H+1}{T} + 6\beta_f \left( \frac{d_{\mathcal{M}}^2 \hat{F}^2 D^2 (H+1)}{4\beta_f^2 T} \right)^{1/3} \right)$$

$$\leq 4\left( \sqrt{\beta_f} d_{\mathcal{M}} \hat{F} D (H+1)^2 T \right)^{2/3} + \tilde{O}(T^{1/2}),$$

and further using Lemma 18 to bound the first term of Theorem 9, and simplifying, we get that

$$\mathcal{R}_{H+1}(T) \leq 12\left( \sqrt{\beta_f} d_{\mathcal{M}} \hat{F} D (H+1)^2 T \right)^{2/3} + \tilde{O}(T^{1/2})$$

$$\leq 12\left( 2 d_x d_u d_{\min} C D_{x,u}^2 \kappa_B \kappa^3 \sqrt{\beta_f / \gamma} (H+1)^3 T \right)^{2/3} + \tilde{O}(T^{1/2}),$$

where the last step only plugs in the values of $d_{\mathcal{M}}, \hat{F}, D$. ∎

## D.1 Proof of Lemma 18

Recall Appendix C.1, where we show that Lemma 12 is a direct corollary of Lemma 15. As the procedure itself does not change here, the proof is concluded by plugging-in our assumptions and parameter choices into Lemma 15. For the first case, $f_t$ are $L$ Lipschitz, and our choice of parameters gives that

$$D_{P_t}^2 = D_{P_1}^2 = D_{P_0}^2 + \frac{4LD\sqrt{T}}{d_{\mathcal{M}} \hat{F}} = H d_{\min}^2 + \frac{4LD\sqrt{T}}{d_{\mathcal{M}} \hat{F}},$$

$$\|P_s^{-1/2}\| \leq r_1^{[1]} \leq \frac{1}{2}\sqrt{\frac{D d_{\mathcal{M}} \hat{F}}{L\sqrt{T}}},$$

and plugging into Lemma 15 we get that

$$\mathcal{R}_1(t) \leq 4\sqrt{d_{\mathcal{M}} L D \hat{F} T^{3/4}} + \underbrace{\frac{H d_{\min}^2}{\eta} + d_{\mathcal{M}} \varepsilon T D_{P_1}}_{R_{\text{low}}},$$

and by our assumptions we indeed get that $R_{\text{low}} \in \tilde{O}(T^{1/4})$ as desired. To conclude the non-smooth case, we further apply Lemma 15 to get that

$$\delta_t = \frac{D}{\sqrt{T}} \qquad \vartheta_t^2 = \frac{d_{\mathcal{M}} D \hat{F}}{4L\sqrt{T}}.$$

Next, for the second case, $f_t$ are $\beta_f$ smooth and our choice of parameters gives that

$$D_{P_t}^2 = D_{P_1}^2 = D_{P_0}^2 + D^2 \left( \frac{4\beta_f^2 T}{d_{\mathcal{M}}^2 \hat{F}^2 D^2} \right)^{1/3} \leq H d_{\min}^2 + \left( \frac{4\beta_f^2 D^4 T}{d_{\mathcal{M}}^2 \hat{F}^2} \right)^{1/3},$$

$$\|P_s^{-1}\| \leq (r_1^{[1]})^2 \leq \left(\frac{d_{\mathcal{M}}^2 \hat{F}^2 D^2}{4\beta_f^2 T}\right)^{1/3},$$

and plugging into Lemma 15 we get that

$$\mathcal{R}_1(t) \leq (4\sqrt{\beta_f}\, d_{\mathcal{M}} \hat{F} D T)^{2/3} + \underbrace{\frac{H d_{\min}^2}{\eta} + d_{\mathcal{M}} \varepsilon T D_{P_1}}_{R_{\text{low}}},$$

and by our assumptions we indeed get that $R_{\text{low}} \in \tilde{O}(T^{1/3})$ as desired. To conclude the smooth case, and thus the proof, we further apply Lemma 15 to get that

$$\delta_t = \frac{D}{\sqrt{T}} \qquad \vartheta_t^2 = \left(\frac{d_{\mathcal{M}}^2 \hat{F}^2 D^2}{4\beta_f^2 T}\right)^{1/3},$$

as desired.