[Reviews · NeurIPS 2020]

Review 1

Summary and Contributions: The paper studies the setting of online linear control with stochastic noise and adversarially chosen costs under bandit feedback. When the costs are smooth and strongly convex the proposed algorithm achieves a root-T regret bound. The online relies on a new way of dealing with bandit convex optimization of losses with

Strengths: The paper makes nice use of the disturbance-action policy reparametrization which reduces the control problem to an OCO problem with bounded-memory losses. The authors then present a new reduction from the no-memory bandit convex optimization to the memory case and show that there is no degradation in performance so long as the costs are smooth (this result may be of independent interest). This continues along the line of work of Anava et al. 2015 which looks at a similar reduction but in the full-feedback setting.

Weaknesses: More practical aspects such as computational complexity considerations were never discussed.

Correctness: The proofs int he main body check out.

Clarity: Some of the definitions are somewhat dense, e.g. Def 5. Just before Def 5, the authors say "...while we do not use it explicitly, we give it here for the sake of correctness." In this case, I would think that including such a definition in the main body would only draw the reader's attention away from the more important aspects of the paper. The same applies in other places of the body where details were perhaps not necessary.

Relation to Prior Work: The related work seems appropriately cited.

Reproducibility: Yes

Additional Feedback:


Review 2

Summary and Contributions: This work proposes a T^1/2 (pseudo)regret algorithm for controlling a stochastic linear system subject to changing sc & smooth loss functions -- the latter may only be accessed as the function value associated with the state-action pair visited. There are two main ingredients here: (1) a perturbation scheme for one-point estimator based on a randomized schedule, (2) a clean, conceptual improved analysis for loss function with memory when using a one-point estimator.

Strengths: + The previously suggested analyses for similar problems do not yield T^1/2, at least in a black-box fashion. For loss function with memory, the regret due to the lag scales as T*(iterate movement). But for bandit problems, the actual (as opposed to expected) iterate movement scales as omega(T^1/2) due to exploration. + In reference to the last point, one innovation here to note that the regret term due to lag scales as T*(expected iterate movement)+T*(exploration)**2 (in lemma 9/ii) due to near-quadratic nature of loss. This is a good conceptual point to make. + The introduction of randomized schedule for perturbations is new to this context.

Weaknesses: + In the constrained case, the use of precond OGD seems essential -- one explores in the ellipsoid of the hessian of a self-concord barrier, which could be quite large in some directions. However, here one can readily play points outside the constraint set. Is the use of preconditioned GD necessary? Note that in context of the previous works it was possible to constrain the disturbance-action class solely in terms of a single frobenius norm, if it is so desired, without degradation of expressivity or the associated regret bound. + Usually in control settings, the reward/cost (even if changing) is often explicitly available, because the algorithm designer herself chooses the cost as a proxy for description of the optimal behavior. At the outset, the reviewer believes, the setting here is a natural extension of the recently studied questions in the community, and completely defensible because research in theory is often forward-looking. Yet, this raises some question on how broadly applicable the setting/results are.

Correctness: The reviewer believes the claim made here are correct.

Clarity: In terms of the technical derivation and comments, this paper is quite well written. The reviewer was follow it linearly, without the need to go back and forth. The prose could at times be more careful (for example, line 166 "O(H) in" -> "O(H) apart in"; there are others).

Relation to Prior Work: Existing algorithms for this setting assume that the entire loss function is available, or that at least a gradient oracle for the same is available (even so if the loss functions are drawn from a distribution). As discussed in "strengths", at least a blackbox adaptation of the previous analyses seems insufficient.

Reproducibility: Yes

Additional Feedback: Thanks for the response -- the score is retained.


Review 3

Summary and Contributions: The paper studies the linear control problem where the dynamics of the system is known, noise is adversarial, costs (convex loss) are adversarial with only bandit feedback observed in each round. This Bandit Linear Control (BLC) problem is reparameterize as a Bandit Convex Optimization (BCO) problem with bounded memory where the cost depends only on last few rounds. The bounded memory BCO is then reduced to a standard BCO with no memory and an optimal algorithm is developed for the BLC when the adversary is restricted to choose smooth cost functions .

Strengths: 1. The reduction of BCO with memory to BCO with no memory is new 2. An optimal algorithm is developed for BLC

Weaknesses: 1. The paper uses too many overloaded notations, which makes reading it hard 2. The connection between the BLC and BCO is not clearly bought out. In BCO the player chooses x_t and adversary draws \xi_t, how is \xi_t related to the cost c_t in the BLC problem Post rebuttal comments: The paper was hard to read. Authors did acknowledge that the notation are too overloded and they had to it to so that they can swithc between the setup. My opinon about the paper remains the same.

Correctness: The results in the appear to be correct. I could not verify the proofs.

Clarity: The presentation can be improved by avoiding overloaded notations and clearly bringing the out the connection between BLC and BCO with bounded memory.

Relation to Prior Work: Relations to prior work cleary discussed.

Reproducibility: Yes

Additional Feedback: There are too many overloaded notaitions, some are listed below. 1. In line 94, \mathcal{K} is defined as collection of matrices. In line 114, it is set of vectors! 2. In line 11, f_t operates on sequence of vectors, in line 140, it operates on sequence on matrices. Other comments: 3. In line 144, the bounds on norm of x_t, u_t, and c_t are explicitly given in terms of problem parameter, then what is the need to assume that they are bounded in line 104? 4. Subsection 2.3 can be shorted to retain only necessary part. With so much (overloaded) notation, it is a distraction. Instead, discussing more about mapping of the BLC to BCO and why the mapping makes sense will be useful. 5. In Algorithm 1, what is \mathcal{S}? 6. Lines starting 193: "...standard arguments apply even if the feedback received by the learner is randomized, as long as it is independent of the learner’s decision on the same round." This statement is not clear. Will the feedback be independent of learner's decision? Then what its use?


Review 4

Summary and Contributions: This paper looks at the problem of bandit feedback in the case of a linear dynamical system with adversarially chosen convex costs. They develop a reduction scheme to convert a Bandit Convex Optimization problem with memory to a BCO problem without memory. They show regret bounds of O(\sqrt{T}) after T rounds for strongly convex and smooth cost functions. They use an approximate reparametrization of online control problem called Disturbance Action policy which casts the problem as bounded memory loss problems. They develop a reduction technique which connects the loss functions with bounded memory to a memoryless version. This is a new contribution in the bandit setting of the problem.

Strengths: This paper looks at the problem of bandit feedback in the case of a linear dynamical system with adversarially chosen convex costs. They develop a reduction scheme to convert a Bandit Convex Optimization problem with memory to a BCO problem without memory. They show regret bounds of O(\sqrt{T}) after T rounds for strongly convex and smooth cost functions. They use an approximate reparametrization of online control problem called Disturbance Action policy which casts the problem as bounded memory loss problems. They develop a reduction technique which connects the loss functions with bounded memory to a memoryless version. This is a new contribution in the bandit setting of the problem. The idea of the paper is easy to follow and seems pretty important to extend the claims of Bandit convex optimization to linear control problems with the reduction from memory to memoryless case under the bandit setting feeling like a significant contribution.

Weaknesses: While the paper's contributions feel pretty strong and it is generally clearly written, some of the math is difficult to follow. In particular, I was not very clear about the disturbance action policy and how generic its reach is, in terms of parametrizing any linear control problem. A separate specific concern was about the evaluation of noise w_t in line 6 of algorithm 1. How is this noise being used in the subsequent updates? Also given the condition in line 8, the updates in lines 9-12 get iterated only a certain number of times in expectation. I missed the connection of how this rolls into the regret rate.

Correctness: I did not go through the math in great detail. In particular, I didn't verify Theorem 9- Lemma 12 at all. The ones prior to that look largely correct.

Clarity: The general idea in the paper is reasonably well described. However given that the main body of the paper is related to new results in bandit feedback problems in linear control, it would be worthwhile to call out which parts of the paper are absolutely novel right at the beginning. These assertions are described throughout the paper and readers might miss out on the main novel contributions unless they go through it in great detail.

Relation to Prior Work: Yes, the first section explains it in reasonable detail. But please refer to the comment above.

Reproducibility: Yes

Additional Feedback: I have read the authors feedback and would like to stick to the score.

[Author Response · NeurIPS 2020]

We thank the reviewers for the time and effort put into the reviews. Below we address the major comments raised in the
reviews; we apologize for brevity due to the space constraints.

**Reviewer 3:**

• **Computational complexity considerations, etc:** Our algorithm is indeed computationally efficient. While this is
mentioned in passing, we agree that this merits an explicit discussion and will add it in the final version. Thanks!
• **Some definitions are dense:** We have given this matter considerable thought while writing. We found that the
definitions given as preliminaries are the minimum requirements for the exposition to be self contained. While it
is possible to state the surrogate costs as an existence result and thus forego the cumbersome details, we thought
this would be too vague for the reader and thus decided to include them in full. In light of your comment, we will
reconsider whether some of the definitions could be deferred or simplified.

**Reviewer 4:**

• **Whether the use of preconditioned GD necessary:** Our methods can indeed apply to an interior point method such
as self-concordant barriers, which would obviate the distinction between the comparator and decision sets. However,
we chose preconditioned GD as it yielded the simplest and cleanest algorithm, while also explicitly showing the
structural differences inside the decision set.
• **Minor comments on writing:** Thank you for pointing these out! We will make the corrections and check for others.

**Reviewer 5:**

Thank you for thoroughly reading the paper and appreciating its contributions! It appears that your main concerns are
around presentation issues; we address these here.

• **Connection between BLC and BCO:** $\xi_t$ in BCO is in fact $w_t$ in BLC. This is mentioned in line 273, which is perhaps
a bit late. We will add a discussion between Sections 4.1 and 4.2 that will explain the relation between BLC and BCO.
• **Overloaded notation:** We used this overloading to name equivalent constructs in the BCO and BLC settings. We
agree that this may cause some confusion and will revise the notations in the final version of the paper. In particular,
to your comments:
1. We will rename the BLC comparator set in line 94, such that it does not clash with the BCO comparator set;
2. $f_t$: We will rename the surrogate costs in line 140, such that they are distinct from the BCO adversarial functions
in line 111.
• **Other responses:**
3. **"In line 144, ... what is the need to assume that they are bounded in line 104?":** In line 104 we assume that a
bound on $x, u$ implies a bound on $c_t$ and its gradient. In line 144 we claim that under the proposed policy $x, u$ are
indeed bounded and thus so is $c_t$.
4. **"Subsection 2.3 can be shorted":** Please see second point in our response to Reviewer 3.
5. **"In Algorithm 1, what is $\mathcal{S}$?":** This denotes sampling from the unit sphere. It is explained in point (2) of the
algorithm description (lines 168-170).
6. **"Will the feedback be independent of learner's decision?":** Our intention was that the feedback is statistically
independent of the player's decision, which in turn means that it provides an unbiased estimate of the desired
underlying adversarial feedback.

**Reviewer 6:**

• **How generic is the Disturbance Action Policy:** This is not a contribution of our work and is discussed in further
detail in previous work (that we deferred details to). In short, the DAP policy can approximate (in terms of cost) any
strongly stable linear policy arbitrarily well (see part (iii) of Lemma 6). We will consider adding an appendix with
further details to make the paper more self contained.
• **Noise in line 6 of algorithm 1 not used in subsequent updates:** The noise $w_t$ is used to calculate the action $u_t$ (in
line 5 of the algorithm). Interestingly, it is otherwise unnecessary for the algorithm, namely, one can separate the
controller from the learning algorithm.
• **How iterating the updates in lines 9-12 rolls into the regret:** This is best seen in Lemma 11 and its proof.
Essentially, lines 9-12 get called about $T/H$ times, incurring the regret of the base algorithm $R(T/H)$. The randomized
schedule prevents the adversary from malicious actions during no update rounds and thus overall the regret is given by
$HR(T/H)$ plus an additional term that accounts for the functions' memory. We appreciate that this is a bit hard to
discern from the current exposition and will attempt to clarify it in the final version.
• **State novelty right at the beginning:** We used the convention that the preliminaries summarize previous work and
the following sections contain only novel contributions unless stated otherwise. Our contributions are summarized in
the introduction but we will make an attempt to distill them to make the distinction even more clear.

[Meta-Review · NeurIPS 2020]

This paper has received positive reviews and is suitable for publication without significant changes. During the discussion, Reviewer #4 pointed out that their question about preconditioned OGD was somewhat misunderstood by the authors: their point was that since the decision set in this setting is unconstrained, there may be no need for preconditioning the gradient updates at all, so omitting this step may make the algorithm and analysis simpler. The authors may consider taking this into account when preparing the final version of the paper.